# NF-κB and Its Role in Checkpoint Control

**DOI:** 10.3390/ijms21113949

**Published:** 2020-05-31

**Authors:** Annika C. Betzler, Marie-Nicole Theodoraki, Patrick J. Schuler, Johannes Döscher, Simon Laban, Thomas K. Hoffmann, Cornelia Brunner

**Affiliations:** Department of Oto-Rhino-Laryngology, Head and Neck Surgery, Ulm University Medical Center, 89075 Ulm, Germany; annika.betzler@uniklinik-ulm.de (A.C.B.); marie-nicole.theodoraki@uniklinik-ulm.de (M.-N.T.); patrick.schuler@uniklinik-ulm.de (P.J.S.); johannes.doescher@uniklinik-ulm.de (J.D.); simon.laban@uniklinik-ulm.de (S.L.); t.hoffmann@uniklinik-ulm.de (T.K.H.)

**Keywords:** NF-κB, immune checkpoint expression, PD-L1

## Abstract

Nuclear factor-κB (NF-κB) has been described as one of the most important molecules linking inflammation to cancer. More recently, it has become clear that NF-κB is also involved in the regulation of immune checkpoint expression. Therapeutic approaches targeting immune checkpoint molecules, enabling the immune system to initiate immune responses against tumor cells, constitute a key breakthrough in cancer treatment. This review discusses recent evidence for an association of NF-κB and immune checkpoint expression and examines the therapeutic potential of inhibitors targeting either NF-κB directly or molecules involved in NF-κB regulation in combination with immune checkpoint blockade.

## 1. Introduction

In recent years, inflammation has been more and more accepted as a hallmark of cancer and is known to play an essential role at all stages of tumorigenesis [1]. Inflammation can contribute to tumor initiation, promotion, metastasis, invasion, and angiogenesis [2]. Nuclear factor-κB (NF-κB), an essential transcription factor necessary for the upregulation of genes important for inflammatory responses, is one of the most important molecules linking inflammation to cancer [3]. The mammalian NF-κB family consists of five members: p50 (NF-κB1), p52 (NF-κB2), RelA (p65), RelB, and c-Rel [4]. NF-κB activation occurs via two signaling pathways, the canonical and the non-canonical pathway. The canonical pathway mediates activation of p50, RelA, and c-Rel, whereas the non-canonical pathway induces p52 and RelB [5]. The canonical pathway is triggered by a variety of inflammatory signals (e.g., proinflammatory cytokines, viruses, Toll-like receptors, antigen-receptors) and leads to rapid and transient NF-κB activation [6,7] Upon stimulation, a trimeric IκB kinase (IKK) complex phosphorylates other IκB family members sequestering NF-κB proteins. Phosphorylation of IκB proteins leads to their ubiquitination and proteasomal degradation, resulting in the release and nuclear translocation of the canonical NF-κB members. In the nucleus, NF-κB proteins can activate genes controlling innate immunity and inflammation [5,7]. In contrast, activation of the non-canonical pathway is slow and persistent and typically induced by ligands of the tumor necrosis factor receptor family [5]. Following stimulation, activation of IKKα leads to phosphorylation and proteasomal degradation of p100, which sequesters RelB. Subsequently, nuclear translocation of RelB and p52 activates genes involved in development of secondary lymphoid organs as well as in B cell maturation and survival [5,7].

NF-κB is activated in cancer cells but also in the tumor microenvironment (TME) of most solid tumors as well as in hematopoietic malignancies. Besides the direct effects of NF-κB in cancer cells—including its modulation of cell cycle genes, apoptosis inhibitors, and invasive proteases—NF-κB also affects gene expression in immune cells, which can result in both promotion and prevention of tumorigenesis [8,9,10]. The role of NF-κB in the anti-tumor immune response is ambivalent and depends on the type of immune cells infiltrating the tumor and also on the TME [2,8]. On the one hand, NF-κB can be activated by cytokines produced by tumor infiltrating immune cells rather than as a result of direct mutations. In turn, NF-κB activates genes controlling several pro-tumorigenic processes such as cell survival, proliferation, growth, angiogenesis, and invasion [2,11,12]. Additionally, NF-κB induces the production of chemokines and cytokines that attract additional immune and inflammatory cells resulting in a positive feed-forward loop to sustain tumor-associated inflammation [13,14]. On the other hand, NF-κB has a key role in the development and function of regulatory T cells (Treg), which have an immunosuppressive function to sustain self-tolerance and immune homeostasis [15,16,17,18]. Treg are also known to suppress anti-tumor immune responses, which explains why NF-κB expression can be associated with the inhibition of anti-tumor immune responses, if the majority of tumor infiltrating cells are regulatory cells [8,19].

Suppression of effector immune functions in the TME is a central mechanism of tumor immune escape. Under physiological conditions the so called immune checkpoints, including programmed cell death protein 1 (PD-1) and T-lymphocyte-associated protein 4 (CTLA-4), are expressed among others on activated T cells. The binding to their ligands PD-L1 or B7, respectively, lead to inhibition of T cell activation, maintaining immune homeostasis and preventing autoimmunity [20,21,22]. Tumor cells can evade the host immune system after upregulation of PD-L1 expression on tumor or infiltrating immune cells in response to inflammatory signals in the TME leading to immunosuppression [22,23,24]. Consequently, the expression of PD-L1 on cancer cells is often associated with poor prognosis [25]. CTLA-4 shares its ligands B7-1 (CD80) and B7-2 (CD86) with the stimulatory receptor CD28. CD28 provides co-stimulatory signals resulting in the activation of transcription factors such as NF-κB, which are required for T cell activation and survival [26]. In contrast, interactions of the ligands with CTLA-4 inhibit T cell responses, but the exact mechanisms are incompletely understood. It is hypothesized that CTLA-4 engagement inhibits NF-κB activation, which might contribute to the downregulation of T cell responses. [27]. Different possibilities have been raised to explain the mechanisms of CTLA-4 function. One model supposes the competition between CTLA-4 and CD28 for ligand binding. Another concept discusses the process of transendocytosis, as CTLA-4 physically captures the B7 ligands resulting in their removal from antigen presenting cells [28,29].

The concept of immune checkpoint inhibition aims to block PD-1/PD-L1 or CTLA4/B7 interactions by using monoclonal antibodies. This leads to the activation of T cells in the TME and finally to the targeting of tumor cells by releasing effector cytokines and cytotoxic granules [30,31,32,33]. The NF-κB signaling pathway is also involved in regulation of immune checkpoint expression in tumor cells, as NF-κB can induce PD-L1 expression thereby promoting T cell suppression and consequently tumorigenesis [34,35,36,37,38,39,40,41,42,43]. Additionally, PD-L1 expression on tumor cells regulates several cell-intrinsic mechanisms promoting tumor cell growth, metastasis, and resistance to Fas-ligand as well as chemotherapy-induced apoptosis [44,45,46]. Moreover, expression of PD-L1 on tumor cells induces their enhanced uptake of glucose from the TME leading to metabolic restriction of T cells, which in turn impairs their anti-tumor immune function and drives tumor progression [47].

This review mainly focusses on the role of NF-κB associated with tumor immune checkpoint expression and examines its therapeutic potential for cancer treatment, particularly in combination with immune checkpoint blockade therapies.

## 2. NF-κB and Tumor Immune Checkpoint Expression

Recently, two independent CRISPR/Cas9-applying studies revealed the NF-κB-dependent signaling pathway among others responsible for the expression of immune evasion genes [48,49]. Among several known immune checkpoint/evasion molecules CTLA-4 and PD-L1 achieved most central clinical relevance for treatment of cancer. The contribution of CTLA-4 and PD-L1 in suppression of immune responses is quite diverse due to the fact that the expression of these molecules is differentially regulated during an ongoing immune response [34]. The expression of PD-L1 depends on various factors. In order to optimize anti-PD-1/PD-L1 therapies, the understanding of PD-L1 controlling mechanisms is highly important. Since the human *PD-L1* promoter encompasses numerous potential NF-κB binding sites, an involvement of NF-κB in *PD-L1* gene regulation was suggested [39,42,50,51]. Indeed, besides epigenetic processes, NF-κB has been shown to regulate transcriptional and posttranslational PD-L1 expression through different mechanisms. NF-κB either directly regulates the expression of the *PD-L1* gene or increases PD-L1 protein expression by enhancing PD-L1 protein stability [19]. By which mechanism NF-κB-mediates PD-L1 upregulation depends on the molecules regulating NF-κB activation. The presence of inflammatory cytokines, like interferon γ (IFNγ), interleukin-17 (IL-17) or tumor necrosis factor α (TNFα), but also oncogenes or tumor suppressors can activate the NF-κB-dependent pathway leading to PD-L1 upregulation and maintenance of immune checkpoint blockade [38,52,53,54,55].

### 2.1. Transcriptional Regulation of PD-L1 Expression by NF-κB

#### 2.1.1. Regulation of PD-L1 Expression by Activation of NF-κB upon Toll-Like Receptor- or Cytokine Receptor-Mediated Signaling

One mechanism of PD-L1 upregulation in immune and cancer cells depends on Toll-like receptor (TLR)-mediated signaling pathways [50,56,57]. Signal transduction via pathogen-associated molecular patterns (PAMPs) and TLRs results in the nuclear translocation of various transcription factors, including NF-κB, and binding of these to the *PD-L1* promoter thereby inducing transcription and translation of *PD-L1* [34]. In solid tumors, upregulated PD-L1 expression via TLR signaling was shown for bladder cancer and gastric cancer [57,58]. A recent study of Li and colleagues reveals that lipopolysaccharide (LPS), a PAMP recognized by TLR-4, increases NF-κB activation, which in turn contributes to PD-L1 upregulation in gastric cancer cells. Furthermore they show that NF-κB regulates *PD-L1* gene transcription through p65-binding to the *PD-L1* promoter thereby increasing PD-L1 expression [58].

Also, IFNs have been shown to regulate PD-L1 expression on tumor and non-tumor cells, whereby IFNγ seems to be the strongest inducer. IFNα was shown to be able to activate PD-L1 expression in hepatocytes, myeloid cells, dendritic cells (DCs), and some cancer cell types in vitro [54,59,60,61]. An involvement of IFNβ signaling was suggested for various cancer cell lines via interferon regulatory factor 9-dependent and independent pathways [62,63]. In addition, IFNβ was reported to enhance PD-L1 expression on monocytes and DCs in vitro and in multiple sclerosis patients in vivo [64]. Although IFNα and IFNβ have been described to activate and signal via the NF-κB pathway, it seems that they mainly induce PD-L1 expression through the Janus kinase (JAK)/signal transducer and activation of transcription (STAT) pathway [65,66]. Studies in dermal fibroblasts revealed that IFNγ induces nuclear translocation of NF-κB thereby increasing *PD-L1* promoter activity and gene expression [67]. Gowrishankar and colleagues additionally showed that IFNγ-inducible expression of PD-L1 is dependent on NF-κB in human melanoma cells. The inducible expression of PD-L1 could be downregulated either pharmacologically using inhibitors of NF-κB signaling or genetically by siRNA mediated NF-κB silencing [37]. However, the exact mechanisms by which IFNγ regulates NF-κB and subsequently PD-L1 remain to be determined. IFNγ was already reported to induce *PD-L1* gene expression via STAT family transcription factors [68,69]. It is described that IFNγ receptor signaling involves STAT transcription factors, which after entry into the nucleus activate transcription of a number of genes. An involvement of STAT3 in PD-L1 upregulation has been reported and there is evidence for crosstalk between STAT3 and NF-κB [14]. Consequently, Gowrishankar et al. investigated an involvement of STAT3 on PD-L1 expression in their study. Inhibition and knockdown of STAT3 had only minor effects on PD-L1 expression suggesting that their observed NF-κB effects were independent of STAT3 [37]. In Epstein–Barr virus (EBV)-positive nasopharyngeal carcinomas PD-L1 expression can be further increased due to the cooperative action of the EBV-associated latent membrane protein 1 (LMP1) and IFNγ. [70]. LMP1 has been described as an activator of the NF-κB pathway [71]. Recently, LMP1 was found to mediate PD-L1 upregulation, which was associated with activation of STAT3, AP-1, and NF-κB [70]. Inhibition of NF-κB effectively suppressed LMP-1 induced expression of PD-L1 in a dose dependent manner in nasopharyngeal carcinoma cells [70]. Moreover, IFNγ upregulated PD-L1 expression in cooperation with LMP-1 [70]. Also in hepatocellular carcinoma (HCC) cells an IFNγ induced PD-L1 expression was observed [53]. In line with Gowrishankar et al., Li and colleagues report that the induction of IFNγ was associated with activation of NF-κB. However, their results rather suggest a strong contribution of the JAK/STAT1 pathway to PD-L1 expression in HCC cells. Furthermore, the authors described a synergistic induction PD-L1 expression by IFNγ together with TNFα. It was postulated that TNFα upregulates expression of IFNγ receptors trough the NF-κB pathway resulting in enhanced IFNγ signaling in HCC cells thereby promoting tumor growth [53].

TNFα was also described to regulate PD-L1 expression trough NF-κB signaling activation in human prostate and colon cancer cells [55]. Since it has been shown that TNFα and IL-17 cooperatively induce expression of downstream genes, Wang and colleagues investigated a possible synergism of TNFα and IL-17 to induce PD-L1 expression [55,72]. However, they observed that both TNFα and IL-17 upregulated PD-L1 via NF-κB but rather individually than cooperatively in human prostate and colon cancer cells [55]. For non-small-cell lung carcinoma (NSCLC), a model in which PD-L1 expression is regulated by DNA methylation and NF-κB during the process of epithelial to mesenchymal transition was described. According to that, PD-L1 expression is controlled simultaneously by DNA methylation and NF-κB signaling. PD-L1 expression required both, demethylation of the *PD-L1* promoter, which can be induced by TGFβ treatment as well as the TNFα-dependent activation of the NF-κB pathway and the subsequent recruitment of NF-κB to the *PD-L1* promoter, in order to promote the expression of the demethylated *PD-L1* promoter [73].

#### 2.1.2. Control of PD-L1 Expression by NF-κB and Oncogene- or Tumor Suppressor Mediated Transcriptional Regulation

After activation of NF-κB p65 via cytokine receptor or TLR-signaling, its binding to specific promoter elements is influenced by many factors. B cell lymphoma 3 (Bcl3), a proto-oncogene and IκB family member is mainly nuclear and contains a transactivation domain. Depending on the composition of NF-κB subunits and transcriptional regulators present at NF-κB responsive promoter elements, Bcl3 activates or represses NF-κB-driven transactivation [74]. Besides hematopoietic malignancies, Bcl3 is upregulated in many types of solid cancers [51,75,76,77,78,79,80,81]. Recently it has been shown that Bcl3 promotes constitutive as well as INFγ-induced PD-L1 expression in ovarian cancer cells. Analyses at molecular level revealed that the *PD-L1* promoter is constitutively occupied by the transcriptional co-activator p300 in ovarian cancer cells [51]. After activation of the IFNγ signaling pathway, Bcl3 expression increases and enables NF-κB p65 acetylation and its p300-dependent recruitment to the *PD-L1* promoter, resulting in enhanced *PD-L1* gene transcription and expression [51]. These findings are in line with previous reports demonstrating a p300-mediated acetylation of NF-κB p65 at Lys-314/315, leading to TNFα-induced NF-κB-dependent gene expression [82,83].

In triple negative breast cancer (TNBC) cells, the upregulation of PD-L1 expression was shown to be Mucin1 (MUC1)-dependent [40]. The oncogene MUC1 is a transmembrane glycoprotein and overexpressed in a variety of tumors of epithelial origin [84]. The MUC1 cytoplasmic domain integrates multiple signaling pathways associated with cancer development and maintenance. Previously, a MUC1-dependent activation of NF-κB was described in several cancer entities. Besides activating signaling pathways leading to the activation and nuclear translocation of NF-κB, MUC1 also directly binds to NF-κB thereby driving the transcription of NF-κB target genes [85,86,87]. In TNBC, MUC1 cytoplasmic domain drives PD-L1 upregulation by Myc- and NF-κB-dependent mechanism [40]. The cytoplasmic domain of MUC1 promotes signaling events finally leading to the direct binding of Myc and NF-κB to the respective binding sites of the *PD-L1* promoter, an essential prerequisite for *PD-L1* gene expression. The authors postulated a potential cross-talk of these two transcription factors [40].

Recently, a previously unrecognized tumor suppressor function of the retinoblastoma protein (RB) was described [42]. Cyclin-dependent kinase (CDK) 4/6-dependent phosphorylation of RB at serine 249 and threonine 252 (S249/T252) facilitates RB to interact with NF-κB p65, leading to inhibition of NF-κB activity and consequently to suppression of *PD-L1* gene expression. In contrast, radiation- or CDK4/6-induced inhibition of RB S249/T252 phosphorylation decreased the association of RB to NF-κB p65 and consequently increased the binding of NF-κB p65 to the *PD-L1* promoter, which further leads to an upregulation of *PD-L1* transcription and expression. Importantly, a transient upregulation of PD-L1 expression after radiotherapy, lasting several days post irradiation was observed [88]. Therefore, the reported radiation-therapy-induced upregulation of PD-L1 mRNA and PD-L1 cell surface expression is possibly mediated by a radiation-dependent decrease of S249/T252 RB phosphorylation, which enables NF-κB p65 dependent *PD-L1* gene transcription [42,89,90,91,92].

#### 2.1.3. Regulation of PD-L1 Expression by NF-κB and Epidermal Growth Factor Receptor Signaling

The relationship between epidermal growth factor receptor (EGFR) mutation and regulation of PD-L1 expression is still controversially discussed, since results of various studies are inconsistent or even contradictory. Thus, several studies suggested an association of PD-L1 overexpression with activating EGFR mutations [93,94,95], while others observed that patients with EGFR mutations have decreased PD-L1 expression or that PD-L1 expression is associated with EGFR wildtype status [96,97,98].

Recently, Guo and colleagues described a stronger expression of PD-L1 expression in EGFR mutant NSCLC cells in comparison to non-mutant EGFR NSCLC, which was associated with increased expression levels of phospho-IκBα and hypoxia-induced factor 1α (HIF-1α) [43]. Therefore, the authors postulated a potential interplay between NF-κB and HIF-1α in the regulation of PD-L1 expression [43]. Indeed, a direct binding of NF-κB to the *HIF-1α* promoter, and vice versa, a HIF-1α-dependent NF-κB activity were reported [99,100,101,102]. However, whether such a mechanism plays a role in the regulation of PD-L1 expression in EGFR mutant cancer cells remains to be elucidated. Lin and colleagues suggested a link between EGFR, NF-κB signaling, and PD-L1 expression. These authors demonstrated an upregulation of NF-κB expression in EGFR-mutant versus EGFR-wildtype cells and an association of EGFR activation with high PD-L1 expression. Moreover, EGFR-tyrosine kinase inhibitors (EGFR-TKI) reduced PD-L1 expression by inhibiting NF-κB in EGFR mutant NSCLC [103]. However, further studies are required to elucidate the interplay between EGFR and NF-κB signaling regarding the regulation of PD-L1 expression.

All of the above-described results suggest that extrinsic stimuli, acting via TLRs or cytokine receptors, but also oncogenes and tumor suppressors either directly activate NF-κB or induce downstream effector molecules finally activating NF-κB. Activation of NF-κB triggers its nuclear translocation enabling its binding to the *PD-L1* promoter. Subsequent NF-κB-mediated transcription and translation of PD-L1 contributes to the observed PD-L1 upregulation and immune evasion in various types of cancer (Figure 1).

### 2.2. Posttranslational Regulation of PD-L1 Expression by NF-κB

Besides its direct effect on the induction of *PD-L1* gene expression by binding to the *PD-L1* promoter, NF-κB can also increase PD-L1 protein expression by enhancing PD-L1 protein stability.

As described in the previous section, EGFR-mediated signal transduction is postulated to be involved in the transcriptional regulation of PD-L1 by NF-κB. Beyond that, EGFR signaling seems to be involved in regulation of posttranslational PD-L1 expression. Upon EGFR stimulation, PD-L1 is glycosylated at four residues resulting in its stabilization in breast cancer cells [104]. In the absence of glycosylation, PD-L1 is phosphorylated by glycogen synthase kinase 3-beta leading to ubiquitination and subsequent degradation [104]. The same group also shows that PD-L1 stability is maintained by the de-ubiquitination of PD-L1 by the fifth element of the COP9 signalosome (CSN5) protein [38]. Mechanistically, in breast cancer cells, Lim and colleagues revealed that TNFα induced NF-κB p65 activation, which in turn binds to the COPS5 gene promoter leading to enhanced transcription of CSN5 having de-ubiquitination activity. Direct binding of CSN5 to PD-L1 leads to removal of PD-L1-bound ubiquitin preventing its proteasomal degradation. The resulting increase in PD-L1 protein stability contributes to tumor immune evasion [38] (Figure 1). Furthermore, in nasopharyngeal carcinoma, a contribution of STAT3 to CSN5 expression was identified and Liu et al. revealed a cooperative binding of NF-κB p65 and STAT3 to the *CSN5* promoter. Mechanistically, CC-chemokine ligand 5, secreted by macrophages, induced formation of NF-κB p65/STAT3 complexes binding to the *CSN5* promoter enhancing *CSN5* transcription leading to PD-L1 de-ubiquitination and stabilization in colorectal cancer cells [105,106]. While transcriptional regulation of PD-L1 via NF-κB has been reported in multiple studies, posttranslational regulation of PD-L1 is poorly understood so far. As outlined above, a few studies provide evidence for NF-κB to regulate PD-L1 protein stability via CSN5, but further studies are required to understand whether NF-κB is involved in other posttranslational regulatory mechanisms of PD-L1 expression.

## 3. NF-κB as Therapeutic Target in Cancer

Because of its key role in tumorigenesis, NF-κB becomes a promising target for cancer therapy. As a consequence, enormous effort has been invested to identify and develop NF-κB pathway inhibitors for cancer treatment. The current NF-κB inhibitors mainly include naturally occurring or synthetic compounds. The major steps of the NF-κB pathway targeted by these compounds include the inhibition of IKK, inhibition of the proteasome as well as the prevention of nuclear translocation of the NF-κB protein and its binding to DNA [107,108]. However, use of these systemic and unspecific NF-κB inhibitors is associated with adverse side effects like systemic inflammation or immunodeficiency due to the pivotal role of NF-κB in both innate and adaptive immunity [109]. In order to minimize systemic toxicity and immunosuppression a more targeted approach including cell-type- and/or subunit-specific inhibition of NF-κB should be considered. NF-κB inhibitors for cancer therapy have been thoroughly reviewed elsewhere [108,110,111]. As outlined in the previous section, NF-κB is involved in the transcriptional and posttranslational regulation of PD-L1 expression. Physiologically, PD-1/PD-L1 signaling negatively regulates T cell mediated immune responses to prevent autoimmunity and to induce peripheral T cell tolerance. PD-1 is predominantly expressed on memory T cells and PD-L1 on various cell types including antigen-presenting cells, T cells, B cells, monocytes, or epithelial cells [112]. Tumor cells can exploit the PD-1/PD-L1 signaling pathway to evade anti-tumor immune responses [113,114]. PD-1 is highly expressed on tumor-infiltrating lymphocytes and tumor cells often upregulate PD-L1 expression facilitating the immunologic response escape [115]. As described in Section 2, PD-L1 expression can be induced by inflammatory cytokines or cancer cell-autonomous mechanisms like mutation dependent oncogenic signaling [116]. Interaction of PD-1/PD-L1 in the TME promotes tumor survival and progression. As a result, PD-L1 expression is generally associated with poor prognosis in numerous malignancies [25,117,118]. Immune checkpoint inhibitors, especially anti-PD-1 or anti-PD-L1 antibodies, aim to disrupt PD-1/PD-L1 signaling in the TME thereby reversing T cell suppression and enhancing anti-tumor immunity [33,112]. Treatment with immune checkpoint inhibitors generated durable responses and extended survival, but not all patients benefit from immune checkpoint therapies.

Since NF-κB can regulate transcriptional and posttranslational PD-L1 expression, as outlined in Section 2, we now focus on the potential of combinatorial treatment of NF-κB inhibitors with immune checkpoint blockade as new promising strategy in order to increase patients’ response rates.

### 3.1. Natural Compounds

There is increasing interest in investigating non-toxic natural compounds with fewer side effects for the treatment of cancer. Several studies indicate that some natural compounds can inhibit NF-κB and might be useful adjuvants for immune-based cancer therapy. One natural compound, which may have the potential to function as an efficient agent to treat cancer associated with inflammation is curcumin. Curcumin, a polyphenol derived from the plant *Curcuma longa*, is known to inhibit the NF-κB signaling pathway by different mechanisms, e.g., by inhibiting IKK activity [119]. Moreover, curcumin monotherapy has already been evaluated for multiple types of cancer showing low toxicity [120]. Additionally, curcumin has also been shown to inhibit CSN5-associated kinase activity [121]. CSN5 deubiquitinates PD-L1 thereby preventing its proteasomal degradation. Inhibition of CSN5 by curcumin was shown to destabilize PD-L1 resulting in diminished PD-L1 expression in various cancer cells thereby enhancing anti-tumor immunity (Figure 2). [38,121]. Furthermore, inhibition of CSN5 by curcumin sensitizes inflammation-induced tumors to anti-CTLA4 therapy in various murine tumor models [38]. A recent study by Xiao et al. demonstrates that combined therapy of NF-κB inhibitor curcumin together with PD-1 blockade significantly improved antitumor immunotherapeutic effect both in vitro and in vivo. The treatment inhibited tumor growth and prolonged survival in a melanoma mouse model [122]. Interestingly, they applied both curcumin and anti-PD-1 monoclonal antibody (mAb) through nanotechnology. Because of its pH sensitivity, the nanodrug is released in the acidic TME. On site, the nanodrug leaves anti-PD-1 mAb to block PD-1 on anti-tumor T cells and generates a new curcumin-encapsulated nanodrug that can be taken up by tumor cells or tumor associated macrophages [122]. The nanodrug has high therapeutic potential since it showed low side effects in vivo and can simultaneously restore tumor killing of cytotoxic T cells and inhibit the NF-κB pathway to recruit anti-tumor T cells into the TME.

Another well-studied natural compound is the grape-derived stilbenoid resveratrol, characterized by anti-inflammatory, immunomodulatory, and anti-cancer properties [127,128]. Using preclinical tumor models, several studies revealed that resveratrol prevents tumorigenesis by modulating multiple pathways dysregulated in cancer [129,130]. Recently, the stilbenoids resveratrol and piceatannol were shown to upregulate PD-L1 expression via HDAC3/p300-mediated NF-κB signaling in breast and colon cancer cells [41]. In this study, treatment with resveratrol or piceatannol resulted in subcellular translocation and nuclear accumulation of NF-κB p65. Furthermore, the increase of PD-L1 expression was attenuated after administration of an IKK inhibitor, suggesting an involvement of NF-κB in the resveratrol- and piceatannol-induced PD-L1 expression [41]. These observations emphasize stilbenoids like resveratrol and piceatannol an interesting therapeutic option to render tumors more sensitive to immune checkpoint inhibitors. On the other hand, prolonged treatment with resveratrol or piceatannol could also promote tumor immune evasion, which underlines the need for more studies especially to determine appropriate dosage levels and treatment periods. Nonetheless, the findings described in this section unveil the co-administration of natural NF-κB inhibitors with checkpoint blockade therapies as a new promising therapeutic option.

### 3.2. Pentoxifylline

Another confirmation for such a combinatorial treatment approach is the finding that inhibitors of the NF-κB subunit c-Rel can potentiate anti-PD-1 therapy. Grinberg-Bleyer et al. show that NF-κB c-Rel regulates Treg function and that c-Rel activity in Treg restricts anti-tumor immune responses. Additionally, they prove that use of the c-Rel inhibitor pentoxifylline (PTXF) in combination with anti-PD-1 antibodies increases the number of tumor infiltrating T cells and potentiates the beneficial effects of PD-1 blockade in a melanoma mouse model [19]. Further advantages of PTXF are its c-Rel specificity, not affecting other NF-κB subunits, and the well tolerability in patients [131]. Inhibition of NF-κB would not only be beneficial regarding PD-L1 expression on tumor cells but also impair the generation and maintenance of tumor associated activated Treg contributing largely to an immunosuppressed TME [19].

### 3.3. TNFα Inhibitors

To increase patients’ response rates to immune checkpoint blockade, the combination therapy of anti-PD-L1 together with anti-CTLA-4 blockers has become a promising treatment approach in recent years, which results in strong and sustained responses [132,133]. However, dual checkpoint inhibition frequently caused autoimmune adverse events in clinical trials [134]. To overcome these side effects and to enhance efficacy of dual checkpoint inhibition, targeting TNFα directly or NF-κB activity could be a promising approach. Initially, it seemed paradoxical that the treatment of patients with anti-TNFα antibodies in order to alleviate the immune-related side effects of a checkpoint immunotherapy led to a general improvement of the treatment response. Immune-related adverse events caused by immune checkpoint blockade are commonly treated with TNF inhibitors [135]. Whether this anti-TNF treatment affects the anti-tumor immune response was relatively unknown. Recently, Perez-Ruiz et al. reported that the prophylactic blockade of TNFα before the start of combined checkpoint inhibition of PD-1 and CTLA-4 can prevent autoimmune adverse effects and additionally enhance anti-tumor treatment efficacy in mouse models [126]. Colorectal cancer mice receiving anti-TNF treatment in addition to double checkpoint blockade had an advantage in tumor rejection and survival compared to mice treated with double checkpoint blockade alone. Moreover, anti-TNF treatment enhanced the effect of double checkpoint blockade leading to increased numbers of tumor infiltrating T cells. In accordance with this study, Bertrand et al. reported a potentiation of anti-PD-1 treatment efficacy by TNF-antibodies and that TNF signaling impairs the accumulation of tumor infiltrating T cells in mouse melanoma [136]. Colitis is one of the most frequent adverse events associated with dual checkpoint inhibition [137]. In a xenograft-versus-host model of colitis double checkpoint inhibition worsened autoimmunity and inflammation, which was markedly reduced by TNF inhibition [126]. Consequently, prophylactic TNF blockade might allow higher doses of checkpoint inhibitors thereby increasing their anti-tumor effects (Figure 2). In addition, Wei et al. demonstrated that macrophage inflammatory responses, including TNFα signaling, elicited NF-κB signaling generating PD-L1^+^ cancer cells. Macrophage depletion inhibited cancer growth mediated by NF-κB signaling and potentiated immune checkpoint blockade [138]. These results suggest a therapeutic strategy combining immune checkpoint treatment and NF-κB inhibition either directly or indirectly by inhibiting TNFα or by depleting macrophages.

### 3.4. Cyclooxygenase 2 Inhibitors

Another approach to target NF-κB for cancer therapy is the use of cyclooxygenase-2 (COX-2) inhibitors. Andrographolide (Andro), a diterpenoid lactone extracted from *Andrographis paniculata*, has been reported to be of therapeutic effect in various cancers showing less adverse effects [139]. In a breast cancer model, Andro has been shown to significantly inhibit acetylation of NF-κB p50 mediated by the transcriptional co-activator p300 histone acetyltransferase. Abrogation of acetylation prevents NF-κB binding to the *COX-2* promotor thereby suppressing its expression. COX-2 suppression by Andro resulted in the inhibition of tumor growth and tumor angiogenesis in a murine breast cancer model (Figure 2) [140]. Celecoxib is another well described specific inhibitor of COX-2, which has also been shown to inhibit IKK activity and to have anti-tumor effects in various human cancers [123,141]. Zuo and colleagues showed that celecoxib inhibits expression of NF-κB in a concentration dependent manner in pancreatic cancer cells. Furthermore, NF-κB inhibition resulted in a distinct reduction of proliferation and invasion of pancreatic cancer cells [142]. COX-2 inhibition has been described in combinatorial therapy with immune checkpoint blockade in several studies. Li and colleagues used an alginate hydrogel system to deliver celecoxib and an anti-PD-1 mAb to tumor local regions [143]. This combination treatment generated potent anti-tumor effects, including inhibited tumor growth, extended survival, and complete tumor regression in both a melanoma and a metastatic breast cancer model. In addition, celecoxib and anti-PD-1 mAb synergistically increased levels of both CD4^+^ INFγ^+^ and CD8^+^ INFγ^+^ T cells within the tumor but also in spleen and draining lymph nodes. At the same time, dual delivery of celecoxib and anti-PD-1 mAb abrogated immunosuppressive mechanisms by reducing Tregs and myeloid derived suppressor cells in the tumor. Furthermore, the combined utilization of these two drugs inhibited angiogenesis and inflammation in their analyzed melanoma mouse model [143]. In line with the above described results, Zelenay et al. also report that celecoxib significantly synergizes with anti-PD-1 treatment in melanoma and colorectal cancer cells. Additionally, they show that anti-PD-1 mAb in combination with aspirin, which blocks both COX-1 and COX-2, is even more potent in promoting rapid tumor regression than with celecoxib [144]. In summary, the combinational therapy of COX inhibitors, suppressing NF-κB expression, and anti-PD-1 mAb simultaneously targets the immunosuppressive and inflammatory TME and might represent a novel anti-cancer treatment option. COX-2 expression might also constitute a useful biomarker of unresponsiveness to immunotherapy and might help to predict treatment outcome.

### 3.5. EGFR-Tyrosine Kinase Inhibitors

As outlined in Section 2, EGFR mutations were reported as genetic drivers of PD-L1 expression thereby contributing to tumor immune escape. [43]. EGFR-TKIs have been identified to reduce PD-L1 expression by inhibiting NF-κB. EGFR-TKIs reduced PD-L1 expression in both EGFR-TKIs sensitive and acquired-resistant NSCLC in vitro and in vivo [103]. EGFR seems to induce PD-L1 expression through the NF-κB signaling pathway, since NF-κB expression in mutant NSCLC was higher compared to wildtype NSCLC. In addition, EGFR-TKI gefitinib reduced PD-L1 expression by attenuating NF-κB activity in tumors [103]. These findings give rise to the speculation that treatment with EGFR-TKIs in combination with PD-1/PD-L1 inhibition might be beneficial for patients with EGFR-TKI acquired-resistance of NSCLC (Figure 2). On the other hand, patients with EGFR-TKI sensitive NSCLC might rather benefit from combinatorial treatment with anti-CTLA-4 antibody, since anti-PD-L1 antibodies can only be efficient in tumor cells with high PD-L1 expression [103]. EGFR-TKIs significantly improve clinical outcome of NSCLC patients with EGFR mutation, but almost all patients develop resistance [145,146,147]. Reports indicate that activation of oncogenic EGFR pathway upregulates PD-L1 expression and enhances susceptibility of tumors to PD-1/PD-L1 blockade treatment in preclinical models [93,148,149]. Since immune checkpoint inhibitors mainly show durable response rates in NSCLC patients, a combinatorial treatment of PD-L1/PD-1 blockade and EGFR-TKIs might be a promising strategy to prolong duration of response and to delay or even prevent resistance [124,125]. Consequently, combination of PD-1/PD-L1 blockade with different EGFR-TKIs was tested in several clinical trials with quite variable results. Some phase III studies could not observe synergistic effects or survival benefits, whereas other early-phase trials reported promising efficacy but at the same time a high incidence of adverse events [125,150,151]. Since the number of patients studied is still limited and only few immune checkpoint inhibitors have been studied in combination with EGFR-TKIs, further studies are required to evaluate whether patients might benefit from this combinational treatment approach.

### 3.6. CDK4/6 Inhibitors

As described in a previous section, CDK4/6 dependent phosphorylation of Rb induces complex formation of Rb and NF-κB p65, thereby inhibiting NF-κB DNA binding blocking PD-L1 expression [42]. Radiotherapy or CDK4/6 inhibitors abolish Rb phosphorylation, with the aim to induce cell cycle arrest in tumor cells, which undesirably also contributes to tumor immune evasion by enhancing PD-L1 expression [42]. Recently, a RB-derived S249/T252 phosphorylation-mimetic peptide, which can overcome undesired tumor immune evasion induced by radiotherapy or CDK4/6 inhibitors, was described [42,152]. Mechanistically, the peptide binds to NF-κB p65 preventing its binding to the *PD-L1* promoter consequently suppressing PD-L1 expression. Co-treatment of the peptide and radiotherapy strongly inhibited tumor growth and increased numbers of tumor-infiltrating T cells [42,152]. Moreover, CDK4/6 inhibitors were found to increase tumor immunogenicity via RB1-dependent mechanisms [153]. CDK4/6 inhibitors increased capacity of tumor cells to present antigen and reduced levels of immunosuppressive Treg in breast cancer models [153]. Findings of Zhang et al. demonstrate that CDK4 regulates PD-L1 stability through cullin 3-SPOP via proteasomal degradation. CDK4 negatively regulates PD-L1 stability by phosphorylating cullin 3-SPOP, an E3 ubiquitin ligase, leading to PD-L1 ubiquitination and degradation [154]. CDK4/6 inhibition increased PD-L1 levels and markedly enhanced tumor regression and improved overall survival in combination with anti-PD-L1 immunotherapy in murine tumor models [154]. Consequently, CDK4/6 inhibitors seem to have the potential to enhance susceptibly of tumors to immune checkpoint blockade.

## 4. Conclusions and Outlook

Although immune checkpoint blockade has yielded promising clinical results, there are still a significant number of tumor patients that do not benefit from such an approach alone. NF-κB has been shown to regulate transcriptional and posttranslational PD-L1 expression thereby contributing to tumor immune evasion. Consequently, NF-κB inhibition can augment immune checkpoint blockade leading to better patient responses. Downregulation of PD-L1 expression on tumor cells with NF-κB inhibitors in combination with anti-PD-1/PD-L1 antibodies could be especially effective to reverse T cell suppression and enhance anti-tumor immunity. The compounds inhibiting the NF-κB pathway described above are featured by low toxicity and selective inhibition of NF-κB signaling components preventing side effects caused by systemic NF-κB inhibition. In addition, inhibitors—such as curcumin, PTXF, or celecoxib—have already been shown to be beneficial for the outcome of immune checkpoint blockade. An additional advantage is that most of the here described compounds are already approved agents and could be further assessed in combination with immune checkpoint blockade in clinical trials. Furthermore, different delivery systems like nanodrugs or hydrogel for application of anti-PD-1 mAb together with NF-κB inhibitors have already been tested. Those tested delivery systems revealed promising results in delivery efficacy and tolerability in preclinical models. Taken together, the current state of research presented in this review suggests co-administration of NF-κB inhibitors together with checkpoint blockade as a new promising approach for cancer treatment. Further investigations will be necessary to evaluate suitable combinations of NF-κB modulators and immune checkpoint inhibitors to get further insights into optimal dosage, treatment schedule, efficacy, and possible adverse events.

## Figures and Tables

**Figure 1 ijms-21-03949-f001:**
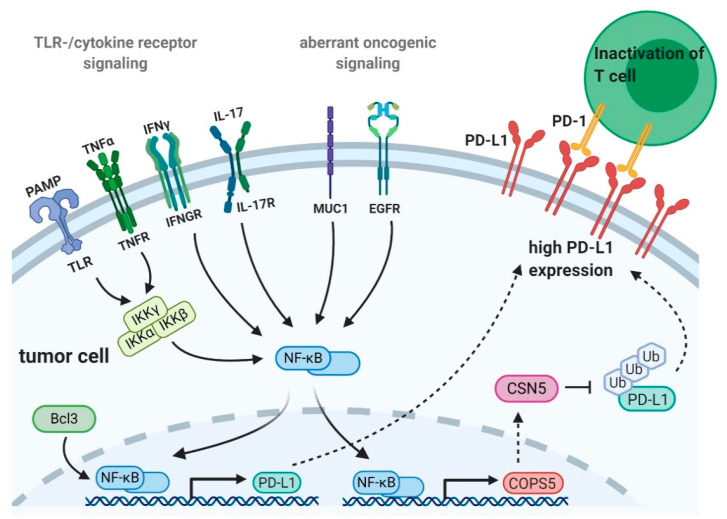
Transcriptional and posttranslational regulation of programmed-death ligand 1 (PD-L1) by Nuclear factor-κB (NF-κB). Nuclear factor κB (NF-κB) is involved in transcriptional and posttranslational regulation of programmed-death ligand 1 (PD-L1) in immune and tumor cells. Toll-like receptor (TLR)- and cytokine receptor-signaling induce NF-κB activation and trigger its nuclear translocation enabling its binding to the PD-L1 promoter [34,55,67]. TLR- and tumor necrosis factor receptor (TNFR)-signaling activate the canonical NF-κB pathway by signaling via the IκB kinase (IKK) complex [7]. The exact mechanisms by which interferon γ (IFNγ) and interleukin-17 (IL-17) activate NF-κB are not completely understood. Aberrant expression of the oncogenes B cell lymphoma 3 (Bcl3) and mucin1 (MUC1) or epidermal growth factor receptor (EGFR) mutations are also described to induce NF-κB-mediated PD-L1 transcription [40,51,103]. NF-κB post-translationally regulates PD-L1 expression by inducing transcription of the COP9 signalosome complex subunit 5 (COPS5) gene encoding the fifth element of the COP9 signalosome (CSN5), which deubiquitinates and therefore stabilizes PD-L1 [38]. All of these mechanisms lead to high PD-L1 expression on tumor cells thereby contributing to tumor immune escape. Arrows indicate paths to NF-κB activation, dotted arrows indicate protein translation and translocation, T-bars indicate inhibition. Figure 1 was created with BioRender.com.

**Figure 2 ijms-21-03949-f002:**
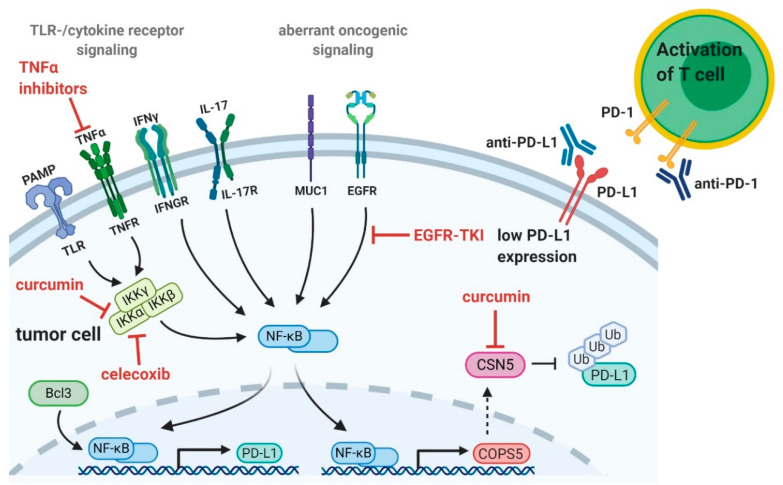
Combinatorial treatment approach of Nuclear factor-κB (NF-κB) inhibition and immune checkpoint blockade. Co-administration of nuclear factor κB (NF-κB) inhibitors and checkpoint blockade seems to be a promising approach to increase cancer patients’ response rates. NF-κB inhibition to reduce programmed-death ligand 1 (PD-L1) expression on tumor cells in combination with anti-PD-1/PD-L1 antibodies could be especially effective to enhance anti-tumor immunity. Curcumin and celecoxib can reduce PD-L1 levels by blocking IκB kinase (IKK) activity and consequently the NF-κB pathway [119,123]. Curcumin can also inhibit the fifth element of the COP9 signalosome (CSN5)-associated kinase activity leading to PD-L1 destabilization [38]. Blocking of oncogenic epidermal growth factor receptor (EGFR) signaling by EGFR-tyrosine kinase inhibitors (EGFR-TKIs) can attenuate NF-κB activity and reduce PD-L1 expression [103]. Combinatorial treatment of EGFR-TKIs and immune checkpoint blockade might prolong duration of response and prevent resistance [124,125]. Tumor necrosis factor α (TNFα) inhibition can reduce adverse events and increase efficacy of immune checkpoint blockade [126]. Arrows indicate paths to NF-κB activation, dotted arrows indicate protein translation and translocation, T-bars indicate inhibition. Figure 2 was created with BioRender.com.

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
