# Peer review of "NF-κB and Its Role in Checkpoint Control"

_ijms, 2020, doi:10.3390/ijms21113949_

Round 1

Reviewer 1 Report

This Review discusses recent findings of interaction between NF-κB and immune checkpoint expression and examines the therapeutic potential of inhibitors targeting molecules involved in NF-κB regulation in combination with immune checkpoint blockade. 

I would recommend to consider this for publishing after revisions and clarifications.

The review must be improved in following aspects:

  1. Activation of nuclear factor κB occurs through two pathways, the canonical and the non-canonical pathway. This information should be inserted in the introduction.
  2. More detailed information about the role of CTLA4 on T cell activation and how it is regulated by NfkB should be included in the introduction.
  3. In section 3, other studies highlighting the potential effect of other natural compounds such as resveratrol on the PD-1 / PD-L1 system should be mentioned.
  4. In 3.5 section, Bibliographical references are few. The authors should mention the results obtained in other studies. Moreover, the potential benefits and problems associated with the combined use of EGFR-TKIs and immunotherapy in patients with NSCLC should be inserted.
  5. Finally, study performed using a combination of CTLA4 and PD-1 check point inhibitors should be inserted

Author Response

Manuscript ID: ijms-755687

Type of manuscript: Review

Title: NF-κB and its role in checkpoint control
Authors: Annika C. Betzler, Marie-Nicole Theodoraki, Patrick J. Schuler, Johannes Döscher, Simon Laban, Thomas K. Hoffmann, Cornelia Brunner

Dear Ms. Aleksandra Redic,

Please find enclosed the revised version of our manuscript by Betzler et al.: "NF-κB and its role in checkpoint control". We are very grateful for the constructive reviewers’ comments and suggestions. We addressed all points made by the reviewers. Finally, the revision improved, to our point of view, very much the quality of our manuscript. We hope that you may now consider the manuscript suitable for publication in the International Journal of Molecular Sciences.

All changes are marked in yellow.

Our revisions included are detailed below in the point-by-point response to the referees’ comments. Once again thank you very much for considering our manuscript for publication in International Journal of Cancer.

Sincerely, yours
Cornelia Brunner, PhD

Point-by-point response to reviewer’s comments.
Reviewer 1: Comments and Suggestions for Authors
This Review discusses recent findings of interaction between NF-κB and immune checkpoint expression and examines the therapeutic potential of inhibitors targeting molecules involved in NF-κB regulation in combination with immune checkpoint blockade.
I would recommend to consider this for publishing after revisions and clarifications.
We are grateful to reviewer 1 for her/his critical reading of our work, and for her/his constructive comments. We performed all suggested changes and included them in the revised manuscript. All changes are marked in yellow.

The review must be improved in following aspects:
1. Activation of nuclear factor κB occurs through two pathways, the canonical and the non-canonical pathway. This information should be inserted in the introduction.
Thank you for this comment. This information is now included.
2. More detailed information about the role of CTLA4 on T cell activation and how it is regulated by NfkB should be included in the introduction.
The relationship between CTLA4 and NF-κB is now discussed in the introduction.
3. In section 3, other studies highlighting the potential effect of other natural compounds such as resveratrol on the PD-1 / PD-L1 system should be mentioned.
This information is now include in section 3 (3.1. natural compounds).
4. In 3.5 section, Bibliographical references are few. The authors should mention the results obtained in other studies. Moreover, the potential benefits and problems associated with the combined use of EGFR-TKIs and immunotherapy in patients with NSCLC should be inserted.
Other studies are cited. Potential benefits and also problems associated with the combined use of EGFR-TKIs and immunotherapy in patients with NSCLC are discussed.
5. Finally, study performed using a combination of CTLA4 and PD-1 check point inhibitors should be inserted
This study was included.

Reviewer 2 Report

Manuscript No. ijms-755687

“NF-kB and its Role in Checkpoint Control” by Annika C. Betzler, Marie-Nicole Theodoraki, Patrick J. Schuler, Johannes Döscher, Simon Laban, Thomas K. Hoffmann and Cornelia Brunner for International Journal of Molecular Sciences

  1. Please also consider the description indicating that the PD-1/PD-L signaling pathway may also serve as a potential mechanism for the escape of cancer cells from immune surveillance. Thus blocking the PD-1/PD-L signaling pathway may have potential therapeutic significance. High PD-L1 expression on tumor cells or infiltrating T cells has also been associated with a more aggressive disease course.
  2. High PD-L1 expression may therefore be a negative prognostic value in oncological patient groups. Please also refer to the such role of ligand for the PD-1 receptor.
  3. The authors basically indicate only IFN-γ as the one that calls up-regulation of the PD-L1 promoter activity and gene expression. Please also indicate in the description type I interferon (α, β), which in vitro also induces PD-L1 expression.
  4. A diagram indicating the place of the described factors in molecular pathways in connection with the immune system and influence on induction/development of tumors would be helpful in understanding the descriptions made.

Author Response

Manuscript ID: ijms-755687

Type of manuscript: Review

Title: NF-κB and its role in checkpoint control
Authors: Annika C. Betzler, Marie-Nicole Theodoraki, Patrick J. Schuler, Johannes Döscher, Simon Laban, Thomas K. Hoffmann, Cornelia Brunner

Dear Ms. Aleksandra Redic,

Please find enclosed the revised version of our manuscript by Betzler et al.: "NF-κB and its role in checkpoint control". We are very grateful for the constructive reviewers’ comments and suggestions. We addressed all points made by the reviewers. Finally, the revision improved, to our point of view, very much the quality of our manuscript. We hope that you may now consider the manuscript suitable for publication in the International Journal of Molecular Sciences.

All changes are marked in yellow.

Our revisions included are detailed below in the point-by-point response to the referees’ comments. Once again thank you very much for considering our manuscript for publication in International Journal of Cancer.

Sincerely, yours
Cornelia Brunner, PhD

Point-by-point response to reviewer’s comments.
Reviewer 2: Comments and Suggestions for Authors

We are very thankful to reviewer 2 for her/his critical reading of our manuscript, and for her/his helpful suggestions. The revised version, according to our point of view, highly support the message of our manuscript. All changes are marked in yellow.

1. Please also consider the description indicating that the PD-1/PD-L signaling pathway may also serve as a potential mechanism for the escape of cancer cells from immune surveillance. Thus blocking the PD-1/PD-L signaling pathway may have potential therapeutic significance. High PD-L1 expression on tumor cells or infiltrating T cells has also been associated with a more aggressive disease course.
Thank you very much for this comment, which we have now considered.
2. High PD-L1 expression may therefore be a negative prognostic value in oncological patient groups. Please also refer to the such role of ligand for the PD-1 receptor.
This information is now included.
3. The authors basically indicate only IFN-γ as the one that calls up-regulation of the PD-L1 promoter activity and gene expression. Please also indicate in the description type I interferon (α, β), which in vitro also induces PD-L1 expression.
Thank you very much for this comment. We have now explained this issue in more detail.
4. A diagram indicating the place of the described factors in molecular pathways in connection with the immune system and influence on induction/development of tumors would be helpful in understanding the descriptions made.
Figure 1 and Figure 2 are now included.

Round 2

Reviewer 1 Report

With the revisions made the review is complete, I also appreciate the inclusion of the figures that certainly enrich the contents.